# Antibiofilm Effect of Curcumin on *Saccharomyces boulardii* during Beer Fermentation and Bottle Aging

**DOI:** 10.3390/biom13091367

**Published:** 2023-09-08

**Authors:** Khosrow Mohammadi, Per Erik Joakim Saris

**Affiliations:** Department of Microbiology, Faculty of Agriculture and Forestry, University of Helsinki, P.O. Box 56, FI-00014 Helsinki, Finland; per.saris@helsinki.fi

**Keywords:** *Saccharomyces boulardii*, curcumin, beer, biofilm, glass

## Abstract

In a prior study, we elucidated the biofilm formation of *Saccharomyces boulardii* on glass surfaces during beer bottle aging. Here, we supplemented brewing wort with curcumin at 25 μg/mL concentration to mitigate *S. boulardii* biofilm and enhance beer’s functional and sensory attributes. An assessment encompassing biofilm growth and development, fermentation performance, *FLO* gene expression, yeast ultrastructure, bioactive content, and consumer acceptance of the beer was conducted throughout fermentation and aging. Crystal violet (CV) and XTT reduction assays unveiled a significant (*p* < 0.05) reduction in biofilm formation and development. Fluorescent staining (FITC-conA) and imaging with confocal laser scanning microscopy provided visual evidence regarding reduced exopolysaccharide content and biofilm thickness. Transcriptional analyses showed that key adhesins (*FLO1*, *FLO5*, *FLO9*, and *FLO10*) were downregulated, whereas *FLO11* expression remained relatively stable. Although there were initial variations in terms of yeast population and fermentation performance, by day 6, the number of *S. boulardii* in the test group had almost reached the level of the control group (8.3 log CFU/mL) and remained stable thereafter. The supplementation of brewing wort with curcumin led to a significant (*p* < 0.05) increase in the beer’s total phenolic and flavonoid content. In conclusion, curcumin shows promising potential for use as an additive in beer, offering potential antibiofilm and health benefits without compromising the beer’s overall characteristics.

## 1. Introduction

Curcumin, a biologically active phytochemical extracted from the traditional spice turmeric, was originally isolated from the rhizomes of the perennial Curcuma longa plant [1]. While curcumin has been widely used in the food industry as a colorant or spice (curcuma), its predominant appeal lies in its pro-health properties. The compound exhibits diverse biological activities including anti-inflammatory, antioxidative, antimicrobial, proapoptotic, and antineoplastic effects, which arise from multiple mechanisms, such as its influence on gene expression and cellular signaling [2,3,4,5]. Curcumin’s impact on glucose metabolism has been observed in studies. It activates p38, a key molecule responsible for glucose uptake in L6 myotubes, in a time-dependent manner [6]. Moreover, curcumin has been shown to reduce blood glucose levels by inhibiting liver gluconeogenesis through affecting the 5 AMP-activated protein kinase (AMPK) signaling pathway [7]. Furthermore, curcumin exhibits antioxidant effects by upregulating the gene expression of superoxide dismutase and glutathione peroxidase [8].

Biofilms are microbial communities that consist of cells encased in an extracellular matrix attached to a surface [9]. The composition of the biofilm matrix can vary among different microorganisms and under diverse growth conditions [10]. However, the biofilm matrix generally comprises extracellular polysaccharides, proteins, and nucleic acids, which provide mechanical stability to biofilms, mediate adhesion to surfaces, and form a cohesive three-dimensional polymer network [11].

Fungi exhibit a remarkable ability to adhere and thrive on various substrates or hosts [12]. For instance, *Saccharomyces cerevisiae* can form biofilms which can be a significant concern in food safety. *S. cerevisiae* has been isolated from biofilms associated with beer bottling plants [13]. Additionally, biofilms can pose significant challenges to public health and clinical settings. For example, biofilm formation on implants and intravascular catheters in medical hygiene represents an important risk factor for bloodstream infections [14]. Several cases of fungemia with *Saccharomyces cerevisiae* var. *boulardii* CNCM I-745 have been reported in critically ill patients [15]. The results from previous studies suggest that *S. cerevisiae* can initiate biofilm formation, which may involve two distinct stages: initial adhesion to the surface and subsequent maturation [16].

Adhesion genes, extensively studied in biofilm formation, enable cells to adhere to surfaces or other cells [17]. *S. cerevisiae* possesses a family of adhesive surface glycoproteins encoded by *FLO1*, *FLO5*, *FLO9*, *FLO10*, and *FLO11* [18,19]. Although they share similar structures with slight differences, these genes serve distinct functions [20]. *FLO1*, *FLO5*, *FLO9*, and *FLO10* facilitate cell–cell adherence and contribute to flocculation [21], while *FLO11* is responsible for cell-surface adhesion, playing a crucial role in agar invasive growth, pseudohyphae, and biofilm formation [22,23]. Different yeast species may have various types of these adhesions [24]. *FLO11* is widely recognized as the key *FLO* gene responsible for surface adhesion, especially in the extensively studied *S. cerevisiae* Σ1278b strain [16,25]. However, the S288c strain has different requirements for biofilm formation, utilizing both *FLO1* and *FLO11*, which are regulated differently [26]. In *S. cerevisiae* wine strains, *FLO5* has been shown to drive adhesive properties related to surface adhesion ability [27]. Several studies have compared biofilm and free cells to comprehend the mechanisms of biofilm formation [28,29].

A previous study revealed that curcumin at a concentration of 50 μg/mL effectively prevented the adhesion of *Candida albicans* to denture materials. Additionally, they found that the inhibitory effect of curcumin was further enhanced when yeast cells were pretreated with curcumin [30]. In light of these findings, our objective was to explore the influence of curcumin on *S. boulardii* biofilm formation on glass surfaces by incorporating it into brewing wort. Furthermore, we assessed the beer’s bioactive content and consumer acceptance through a sensorial test.

## 2. Materials and Methods

### 2.1. Beer Brewing Process

A total of 5 kg of malt blends containing 2.5 kg of Pilsner malt, 2 kg of Pale ale malt, and 0.5 kg of wheat malt (Viking malt, Lahden polttimo Ltd., Lahti, Finland) were used for beer brewing according to the previously optimized procedure [31].

The brewing process was as follows:i.Malt crushing: the malt was mixed and ground with a two-roller mill at a setting of 0.5 mm.ii.Malt mashing: mashing was carried out in a 27 L capacity Brewferm^®^ (Sunderland, UK) stainless steel boiler equipped with a mash agitator with a water ratio of 1:4 for one hour at 66 °C.iii.Lautering: a mash filter was used for lautering, and the mash was washed with 70 °C water until the Brix value reached 5° Bx.iv.Wort boiling: the wort was boiled for one hour; bittering hops (10 g Columbus hops) were added at the beginning of the boil, and aroma hops (30 g Hüll Melon) were added 10 min before the end.v.Beer fermentation: the wort was pitched with *S. boulardii* (6.2 × 10^5^ cells/mL) in a total volume of 10 L in each flask. Prior to treatment, curcumin (Sigma cat # C1386) was dissolved in alcohol (99.8%) to obtain a stock solution (3 mg/mL) and used for treatment at a concentration of 25 μg/mL. Fermentation was also conducted in 250 mL bottles to investigate the impact of curcumin on fermentation performance and biofilm development. A control sample was prepared without curcumin addition. All samples were fermented in the dark at 20 °C for two weeks. During this period, they were regularly tested every two days to monitor fermentation performance and biofilm development.vi.Beer aging: Beer samples were supplemented with d-glucose (5 g/L) and subsequently aged for a duration of four weeks in 250 mL bottles. The bottles were stored in a dark environment at a temperature of 20 °C.

### 2.2. Biofilm Surface and Experiment Time Points

Biofilm formation was conducted using an eight-chambered glass slide (Thermo Fisher Scientific, Vantaa, Finland) and a glass cover slide measuring 18 by 18 mm. These slides were submerged in beer bottles to enable examination through biofilm quantitation assays and confocal laser scanning microscopy (CLSM). Three different bottles from the same batch were analyzed at each time point. Biofilm analysis was carried out during weeks 1 and 2 (fermentation period) and weeks 3, 4, 5, and 6 (bottle aging period). Samples intended for cell count and sugar consumption analysis were examined every other day over a span of two weeks. These samples were degassed using an ultrasonic water bath and subsequently filtered through a plain disc filter paper (Whatman, Maidstone, UK, 11 μm) and a cellulose acetate membrane syringe filter unit (pore size of 0.22 μm, Bionordika, Solna, Sweden).

### 2.3. Quantification of Biofilm Level

#### 2.3.1. Crystal Violet Staining Assay

The biofilms’ biomass was assessed using an indirect method adapted from the technique described by Kovač et al. [32]. The eight-chambered glass slide was subjected to a wash with phosphate-buffered saline (PBS) to remove nonadherent cells and air-dried for biofilm quantification. The biofilm mass was fixed using 200 μL of 99% methanol for 15 min. Subsequently, the wells were stained with 200 μL of 0.5% crystal violet solution and incubated at room temperature for 20 min. After the incubation, the eight-chambered glass slides were washed with sterile distilled water to remove excess stain. The wells were then destained by adding 200 μL of 33% (*v*/*v*) acetic acid for 5 min. To quantify the biofilm, 100 μL of the acetic acid from each well was pipetted into a 96-well plate, and a spectrophotometer (infinite M200, Tecan, Grödig, Austria) was utilized to measure the absorbance at 595 nm. The average absorbance of the control wells was subtracted from the absorbance of the sample wells, and the resulting calculated averages were graphed, along with standard deviations.

#### 2.3.2. Biofilm XTT Reduction Assay

XTT assay was used to evaluate the viability of yeast cells within the biofilm based on the reduction of a tetrazolium salt [33]. An amount of 200 μL of the XTT/menadione solution (4 mg XTT was dissolved in 10 mL prewarmed PBS and supplemented with 100 μL of menadione stock solution (Sigma-Aldrich, Saint Louis, MO, USA), which contained 55 mg of menadione in 100 mL of acetone) was added to each well and incubated in the dark at 37 °C for three hours. Then, the suspension was agitated, transferred to microcentrifuge tubes, and centrifuged (4000× *g*, 5 min) to pellet cells. The optical density of the supernatant (formazan) was measured at 492 nm.

### 2.4. Cell Count

Viable cell counts for pitched yeast, total, and cells in suspension during beer fermenting and aging were evaluated by standard spread plate technique. Cell count was monitored every 2 d during fermenting (up to 14 d) and every week during aging until one month. Briefly, the head pressure in the beer bottle was released very slowly to prevent foam formation and affecting the cells in suspension. Using a volumetric pipette, 10 mL of beer was taken from the middle of the bottle and dispersed in 90 mL dilution medium (containing 8.5 g/L NaCl, 1 g/L peptone, 0.3 g/L Na_2_HPO_4_·2H_2_O, pH 5.6). In the case of total cell count, bottles were inverted several times after degassing in an ultrasonic water bath (Sonorex Super 10P. Bandelin, Berlin, Germany) to ensure an even distribution of yeast cells. After preparing serial dilutions, the spread plate technique was used for culturing yeast cells on Sabouraud chloramphenicol medium. Following incubation at 28 °C for 72 h, colony identity was confirmed on the medium through PCR amplification of the internal transcribed spacer (ITS) region using universal fungal primers ITS1 (5′-TCCGTAGGTGAACCTGCGG-3′) and ITS4 (5′-TCCTCCGCTTATTGATATGC-3′) applied to the colony DNA [34].

### 2.5. Physicochemical Analysis of Wort and Beer

#### 2.5.1. HPLC Analysis

Sugar concentration (glucose and maltose) and produced ethanol were measured by an Alliance high-performance liquid chromatography system (Waters, Milford, MA, USA, Separation module e2695) coupled with autosampler and two consecutive detectors, Waters 996 photodiode array, and Hewlett Packard HP1047A RI. Empower 3.0 software was used for data collection and analysis. The analytes were separated on Hi-Plex H, 300 × 6.5 mm (Agilent technologies, Santa Clara, CA, USA) column at a flow rate of 0.5 mL/min at 40 °C. The running buffer was H_2_SO_4_ (5 mM). All beers were run through filters (Whatman, TishScientific, North Bend, OH, USA) with a pore size of 0.45 µm before being injected into the instrument.

#### 2.5.2. pH Measurement

The pH tests were in accordance with the AOAC Method 945.10 (2012) for beer, using a pH/reference electrode system.

### 2.6. Reverse Transcription Quantitative PCR (RT-qPCR) Analysis of FLO Genes

Yeast cells were lysed using glass beads and a bead beater (FastPrep-24, MP Biomedicals) at stationary phase of growth. RNA purification reagents and initial protocols were based on the RiboPure™-Yeast kit (Ambion, Applied Biosystems, Austin, TX, USA). Then, the extracted RNA was treated with ultrapure DNase I to eliminate genomic DNA contamination. Spectrophotometric measurements confirmed the quantity and quality of purified RNA at 260 and 280 nm and agarose gel electrophoresis. RNA sample (1 μg/μL) was reverse transcribed into cDNA using RevertAid H Minus First Strand cDNA Synthesis Kit. We evaluated mRNA levels between treated/untreated cells in comparison to a housekeeping gene, 18S. For this purpose, we designed primers for the cDNA sequences and evaluated five genes involved in flocculation and biofilm formation (*FLO1*, *FLO5*, *FLO9*, *FLO10*, and *FLO11*) (Table 1).

qRT-PCRs were performed on cDNA samples in a final volume of 25 μL, including 2 μL cDNA (100 ng), 12.5 μL 2× SYBR Green/ROX Master Mix (Thermo Fisher Scientific), 0.5 μL forward and reverse primers (0.3 μM), and 9.5 μL sterile ddH_2_O. Real-time PCRs were performed in triplicate in an Applied Biosystems™ QuantStudio™ 3D digital PCR system and a negative control was used to ensure the absence of nonspecific amplification. The amplification cycles were 25 cycles of 95 °C for 15 s, 60 °C for 60 s, and 72 °C for 30 s, followed by an extension at 72 °C for 5 min. The obtained results were analyzed using the comparative critical threshold (2^−ΔΔCt^) method and reported as n-fold differences compared to untreated cells.

### 2.7. Microscopic Techniques

#### 2.7.1. Confocal Laser Scanning Microscopy (CLSM) Observation

The effectiveness of curcumin on biofilm formation was quantitatively analyzed and visualized using CLSM. On weeks 1, 2, and 6, the cover glasses were removed from beer bottles and gently washed with sterile PBS. Later, cover glasses were stained with fluorescein isothiocyanate (FITC)-conjugated concanavalin A (ConA) (50 µg/mL) and prepared with 10 mmol^− l^ of HEPES buffer solution for 30 min at room temperature. The excess stains from the cover glasses were removed with subsequent washes with PBS, were placed on slides with the biofilm pointing up, and were imaged under CLSM (Leica Microsystems GmbH, Wetzlar, Germany) with excitation wavelengths of 495 and 525 nm, operating at a magnification of ×10. Z-stacks were taken with an interval of 0.2 μm and analyzed by IMARIS software (version 10).

#### 2.7.2. Transmission Electron Microscopy (TEM) Analysis

The effect of curcumin on the ultrastructure of *S. boulardii* cells was observed under TEM. Cells were prefixed in a mixture of 3% paraformaldehyde and 0.5% glutaraldehyde (in 0.1 M Na-citrate buffer, pH 7.2) for 90 min at room temperature. Then, the cells were washed in Na-citrate buffer three times. Subsequently, the samples were fixed in 2% osmium tetroxide (OsO_4_) for 2 h at 4 °C. Fixed cells were washed in phosphate buffer for 30 min, dehydrated in a graded series of ethanol (70 ± 100%), and embedded in low-viscosity resin. Ultrathin sections (80 nm) were stained with uranyl acetate (UAc) and lead citrate (LC) and viewed with a Jeol JEM-1400 electron microscope. Digital images were collected using a Gatan Orius SC 1000B bottom-mounted CCD camera (Gatan Inc., Pleasanton, CA, USA).

### 2.8. Determination of Total Phenolics

The total phenolic content (TPC) in beer samples was determined using the Folin–Ciocalteu spectrophotometric method, as described by Zhao et al. [35]. Briefly, 0.5 mL of diluted beer sample was mixed with 2.5 mL of a 10-fold diluted Folin–Ciocalteu’s phenol reagent and incubated for 5 min. Subsequently, 2 mL of a 7.5% sodium carbonate solution was added, and the final volume was adjusted to 10 mL with deionized water. The solution was then shaken, and after 1 h of reaction at room temperature, the absorbance at 760 nm was measured using a spectrophotometer (infinite M200, Tecan, Austria). A calibration curve was prepared using a gallic acid solution in the concentration range of 0.01 to 1 mg/mL, and the absorbance readings of the beer samples were compared to this calibration standard curve. The results were expressed as milligrams of gallic acid equivalents per liter of sample (mg GAE/L).

### 2.9. Sensorial Analysis

After four weeks of bottle aging, the beers were subjected to evaluation using the author’s questionnaire. The panelists were provided with two coded samples of beer (w/o CUR) to assess various sensory aspects including appearance, aroma, taste, mouthfeel, aftertaste, and overall impression. A panel comprising 18 trained testers carried out the evaluations, utilizing a hedonic scale ranging from 1 to 9, where 1 indicated “extremely dislike” and 9 represented “extremely like”.

### 2.10. Statistical Analysis

The effect of curcumin on *S. boulardii* biofilm formation was evaluated by one-way analysis of variance, followed by Tukey’s multiple comparison test. Statistical analysis was computed by using SPSS 22.0. Data were expressed as the mean ± SD. In all analyses, *p*-values of 0.05 or less were considered statistically significant.

## 3. Results

### 3.1. Curcumin as a Potential Inhibitor of Biofilm Formation in S. boulardii

The total biofilm biomass and yeast cell viability were assessed within the biofilm using CV staining and the XTT reduction assay, respectively. The test involved introducing curcumin to one set of wort while the other set of wort served as a control with an equivalent concentration of sugars. This allowed for a comparative analysis under identical conditions. The experiments were conducted at six specific time points: weeks 1 and 2 during the fermentation phase as well as weeks 3, 4, 5, and 6 during the bottle aging period. Our results revealed that the addition of curcumin to the brewing wort considerably reduced both biofilm mass (Figure 1A) and metabolic activity of *S. boulardii* (Figure 1B).

The initial adhesion process directly impacts the subsequent biofilm formation and development. In the first week of fermentation, the beer supplemented with curcumin (B/CUR) exhibited a CV absorbance value of 0.07, indicating limited biofilm formation. In contrast, the control sample had already developed a more substantial biofilm, as reflected by a total biomass measurement of 0.13. This disparity highlights the inhibitory effect of curcumin on the early stages of biofilm formation. Moreover, curcumin significantly influenced the development of a preformed biofilm at the end of fermentation (*p* < 0.05). The CV absorbance value of the B/CUR at week 2 was recorded as 0.09, which was significantly lower compared to the control beer’s value of 0.18. These results indicate that curcumin affects both the initial formation and subsequent development of biofilm by *S. boulardii.* During beer bottle aging, the establishment and development of the biofilm exhibited a progressively increasing trend; however, notable variations were observed in comparison to the control group (Figure 1A).

The XTT reduction assay yielded similar results. During the fermentation and aging period, the XTT absorbance values obtained from the control group were significantly higher (*p* < 0.05) than the B/CUR samples (Figure 1B). Consequently, it can be inferred that the addition of curcumin resulted in a reduction in biofilm formation and development in contrast to the control group.

### 3.2. Influence of Curcumin on Fermentation Performance

A fermentation experiment was performed to evaluate the fermentative behavior of *S. boulardii.* The study focused on assessing parameters such as viable cell count, sugar uptake, and ethanol production rate. At the early stage of fermentation, curcumin supplementation considerably affected the yeast’s fermentation performance. During the initial four days of fermentation, the growth and sugar uptake rate of yeast in B/CUR were significantly (*p* < 0.05) lower than those of controls (Figure 2). On the second day of fermentation, the total number of yeast cells increased from 5.8 to 8.4 log CFU/mL. However, in B/CUR, cell growth exhibited a significant difference (*p* < 0.05) and reached 6.5 log CFU/mL. Despite the initial difference in yeast population at the beginning of fermentation, by day 6, *S. boulardii* developed an adaptive response; the number of cells in the B/CUR nearly equaled that of the control beer (8.3 log CFU/mL), and it remained stable thereafter. However, throughout the entire fermentation period, the B/CUR showed fewer cells in suspension than the control group. As a result, the supernatant of the B/CUR samples displayed a significantly clearer appearance than the control beer. The total number of yeast in both the control and test samples gradually decreased throughout the aging period, reaching 7.1 and 7.5 log CFU/mL, respectively, at the end of the aging period.

Figure 3 shows the uptake order of wort sugars and ethanol production by *S. boulardii* in beers. The initial concentration of sugars in the pitching wort was as follows: maltose (92.2 g/L), glucose (8.8 g/L), fructose (1.75 g/L), and sucrose (1.4 g/L). In control beer, glucose and sucrose consumption occurred during the first two days of fermentation. From the fourth day of fermentation onwards, the glucose content remained approximately 0.5 g/L. On the second day of fermentation, there was a significant difference (*p* < 0.05) in glucose consumption between B/CUR (2.5 g/L) and the control beer (1 g/L); however, on the fourth day of B/CUR fermentation, the glucose concentration was almost the same as in the control beer. This difference was more evident in the case of maltose hydrolysis. On the second and fourth days of fermentation in B/CUR, the concentration of maltose was 77.4 g/L and 48.4 g/L, respectively, compared to 59.6 g/L and 17.3 g/L in the control beer.

Consequently, there was a direct correlation between sugar consumption and ethanol production. Ethanol production increased as the yeast cells metabolized the fermentable sugars in the brewing wort. Most ethanol production occurred on the fourth day of fermentation, but the ethanol production rate in B/CUR samples was slower than the control. During the four-day fermentation period, when the cell growth reached a value of 7.8 log CFU/mL, and an average of 53% of the total fermentable sugars were consumed, a total of 33 g/L of ethanol was produced. However, measurements taken after day 6 of fermentation showed insignificant differences (*p* > 0.05) compared to the control assay. Finally, at the end of fermentation, yeast cells actively consumed fermentable sugars in the brewing wort, producing up to 54 g/L of ethanol. This finding suggests that the fermentation process had stabilized, and further changes in yeast population and sugar consumption were not significantly (*p* < 0.05) different from the control group.

### 3.3. Impact of Curcumin on Transcriptional Changes in FLO Genes

To further explore the mechanism by which curcumin mediates its effects on *S. boulardii* biofilm formation, qRT-PCR was employed to assess the gene expression levels within the *FLO* gene family (Figure 4). Gene expressions in the control group were defined as 100%. In contrast, expressions in the B/CUR were calculated as relative values to assess the impact of curcumin on biofilm formation. Curcumin treatment caused a significant decrease in the expression of the *FLO* gene family, except for *FLO11*, which showed a non-significant increase. The expression levels of *FLO1*, *FLO5*, *FLO9*, and *FLO10* genes were significantly downregulated by 18.5-fold, 16.1-fold, 6.5-fold, and 15.5-fold, respectively. In contrast, the expression of the *FLO11* gene was only slightly upregulated by 0.5-fold. These findings indicate a significant decrease in flocculation ability and biofilm formation, suggesting that the downregulation of these genes likely contributes to the observed effects.

### 3.4. Prevention of Biofilm Formation on Glass Microscope Slide Covers

Prevention of biofilm formation and development by curcumin was confirmed by fluorescence microscopic examination. We depicted biofilms formed on cover glass submerged in beer bottles with and without curcumin, focusing on the exopolysaccharide (EPS) content of the extracellular matrix. We used FITC-conA dye to stain the extracellular polysaccharides and measure biofilm thickness (Z-stack images) by confocal laser scanning microscopy. Figure 5 shows a fluorescent microscopy comparison of biofilms formed by *S. boulardii* in both control beer and B/CUR during fermentation (weeks 1 and 2) and the last week of bottle aging (week 6). Figure 5 shows a substantial reduction in biofilm formation in the presence of curcumin. During the first week of fermentation, wort supplementation with curcumin led to minimal, if any, visible cells adhering to the surface of the glass, lacking the three-dimensional biofilm structure, whereas a significant biofilm was formed in the control group. As the fermentation progressed, *S. boulardii* developed large aggregates that are characteristic of mature biofilms on the submerged cover glass surfaces. However, the formation of these aggregates was reduced in curcumin-treated samples. At the end of beer bottle aging, the biofilms grown in B/CUR exhibited a significantly reduced thickness, ranging from 14 to 26 μm, compared to the biofilms grown in the control group, which had a thickness ranging from 24 to 38 μm. This finding demonstrates that curcumin at a concentration of 25 μg/mL significantly affected biofilm formation and development of *S. boulardii* on the cover glass substrate.

### 3.5. Effect of Curcumin on the Ultrastructure of S. boulardii

The observation under TEM demonstrated that curcumin at a relatively low concentration of 25 μg/mL did not induce dramatic changes in the *S. boulardii* ultrastructure. The *S. boulardii* cells showed a homogeneous cytoplasm surrounded by a smooth cell membrane and regular cell wall. The endoplasmic reticulum was distributed along the inner side of the plasma membrane.

We noticed that high concentrations of curcumin induced dramatic ultrastructure changes. In the presence of ½ MIC curcumin, the ultrastructure of the cells had drastically changed, characterized by a greater number of irregularities in the cell wall and inner plasma membrane. The outer layer of the cell wall displayed increased electron density (arrow in Figure 6C). Additionally, the cell walls appeared deformed, and alterations were observed in the cytoplasmic membrane. Furthermore, the fragmented separation between the cell wall and the plasma membrane was observed (arrow in Figure 6D).

### 3.6. Evolution of pH and Bioactive Compounds

Table 2 shows the pH and polyphenol content values recovered after beer fermentation and bottle aging. In beer production, pH plays a crucial role, significantly influencing yeast behavior and the synthesis of various metabolites. The pH value dropped substantially during the first four days of fermentation. However, after this period, minimal variations in pH were noted throughout the fermentation and bottle aging period. The pH values, recorded in all cases at the end of the bottle aging (about 4.5), are characteristic of a beer. Only minimal differences were recorded between the B/CUR and control beer.

At the end of the fermentation period, the B/CUR exhibited a total phenolic content of 283 ± 22 mg GAE/L and a total flavonoid content of 23.6 ± 2 mg QE/L. These values were significantly (*p* < 0.05) higher compared to the control beer, which had a total phenolic content of 229 ± 18 mg GAE/L and a total flavonoid content of 15.2 ± 2.6 mg QE/L (Table 2). However, it is worth noting that the concentration of bioactive compounds (TPC and TFC) decreased during the beer bottle aging phase. Nonetheless, this reduction in bioactive content was found to be statistically insignificant (*p* > 0.05).

### 3.7. Sensorial Analysis of the Beers

The beers underwent sensory analysis following the maturation process to assess consumer acceptance. The acceptability of the beers was evaluated based on six sensory parameters: appearance, aroma, taste, mouthfeel, aftertaste, finish, and overall impression. The tasters provided grades for each sample using a nine-point hedonic scale, as shown in Table 3. The evaluation conducted by the tasters revealed a significant difference (*p* < 0.05) among the samples with regard to the assessed attributes. The attribute appearance was the one with the highest mark, with 8.2 for the beer supplemented with curcumin. Concerning taste, aftertaste, and finish, the tasters awarded higher marks of 7.7 and 7.3 for the beers produced with curcumin compared to 7.1 and 6.6 for the control beer, respectively. However, the two groups had no significant difference in aroma and mouthfeel attributes. According to the results of the acceptance test for the attribute “overall impression”, 27.7% of the tasters rated the beer produced with curcumin as “like extremely” (score 9), 55.5% as “like very much” (score 8), and 16.6% as “like moderately” (score 7).

On the other hand, for the control beer, 11.1% of the tasters marked it with a “like extremely” (score 9), 38.8% with “like very much” (score 8), and 50% with “like moderately” (score 7). These findings indicate that the beer produced with curcumin received higher ratings for overall impression, with a larger percentage of tasters showing stronger positive preferences (“like very much” and “like extremely”) compared to the control beer (Table 3). Based on these ratings, both beer samples were accepted by the tasters. However, the beer produced with curcumin received higher marks, indicating higher sensorial acceptability among the tasters.

## 4. Discussion

In our previous work, we reported the biofilm formation of *S. boulardii* on glass surfaces during beer bottle aging [31]. In this research, we employed curcumin as a natural compound to prevent biofilm formation of *S. boulardii* biofilm on glass surfaces. During our pilot studies, we observed that curcumin’s effects on the growth rate and ultrastructure of *S. boulardii* were dose-dependent. Notably, we determined the minimum inhibitory concentration (MIC) value of curcumin for the planktonic cells of *S. boulardii* to be 200 μg/mL. Based on these findings, our investigation focused on evaluating the effects of an optimum concentration, specifically, 25 μg/mL of curcumin. The findings from our study were quite surprising. As seen in Figure 1, curcumin supplementation caused a marked decrease in the level of biofilm measured by crystal violet staining and XTT reduction methods. We further support this statement by fluorescence staining of extracellular polysaccharides and monitoring the biofilm formation and progress. Cell-surface adhesion is the initial and crucial step in the process of biofilm formation [16]. The secretion of extracellular polysaccharides (encoded by *FLO1* gene) plays a crucial role in the development of complex biofilm structures [36,37]. The data obtained from confocal laser scanning microscopy (CLSM) images revealed that adding curcumin into the brewing wort reduced adhesion and biofilm development compared to the control group.

Environmental variations, particularly stress conditions and nutrient scarcity, can alter the cell wall’s adhesion properties [38,39]. These modifications manifest in various phenotypes, including flocculation, biofilm formation, or substrate adhesion [16,40]. To elucidate the impact of curcumin on the expression of *FLO* genes, we quantified *FLO* gene transcription levels using qRT-PCR. Compared to the control *S. boulardii*, the adhesion genes *FLO1*, *FLO5*, *FLO9*, and *FLO10* exhibited reduced expression in the yeast cell transcriptome during the stationary phase. The gene *FLO11*, responsible for encoding a flocculation protein involved in cell–substrate adhesion, plays a crucial role in biofilm formation [26,41]. However, based on qRT-PCR analysis, it was found that the expression of *FLO11* in the B/CUR remained nearly unchanged. Consequently, *FLO11* was ruled out as the primary cause of the reduction in biofilm formation in B/CUR. This result suggests that the decline in biofilm formation in B/CUR is mainly attributed to the reduced expression of these flocculation genes. In a study conducted by Zhang et al. [42], it was observed that the deletion of ARO8 and ARO9 genes resulted in reduced levels of 2-PE, leading to a subsequent decrease in biofilm formation at the early fermentation stage. Through qRT-PCR analysis, they found that the expression of *FLO11* in ΔARO8 and ΔARO9 was downregulated by less than twofold, while the expression in ΔARO10 remained mostly unchanged. These results indicated that *FLO11* was not the primary factor responsible for the reduction in biofilm formation in ΔARO8 and ΔARO9 mutants.

Our data revealed that adding curcumin to the brewing wort reduced *S. boulardii* growth and fermentation performance immediately after pitching. However, as time progressed, the strain *S. boulardii* exhibited comparable growth and fermentation performance in the presence of curcumin. The supplementation of beer with curcumin did not significantly impact the yeast’s overall growth and its ability to carry out the fermentation process. These findings align with previous research using yeast as a model organism, demonstrating the sensitivity of *S. cerevisiae* to curcumin through diverse pathways and mechanisms [43,44]. An investigation carried out by Minear et al. [45] showed that curcumin acts as an effective chelator of Fe(III) and consequently inhibits the growth of *S. cerevisiae*. Their research demonstrated that curcumin-induced growth inhibition is dose-dependent. Low iron level affects the DNA repair activity of Fe(II)/2OG-dependent dioxygenases [46]. A recent study by Stępień et al. [47] explored the effects of curcumin on both replicative and chronological aging in different *S. cerevisiae* strains. Across all analyzed strains, curcumin supplementation at concentrations of 200 μM and 300 μM was found to be a stress-inducing factor, led to a clear inhibition of the growth rate, and extended the mean doubling time.

The utilization of an optimized concentration of curcumin can enhance the quality of the beer. Our research also revealed curcumin’s favorable impact on the sensory characteristics of the beer. The tasting panel clearly identified B/CUR as having superior qualities. The presence of iron and heavy metals in craft beers in quantities surpassing the recommended exposure limits set by sanitary standards can pose a risk to human health. Furthermore, such elevated levels of heavy metals can have a detrimental impact on the beer’s quality and stability [48]. It is noteworthy that iron, in particular, has the potential to generate off-flavors and induce color change in beer by interacting with polyphenols [49,50].

Previous studies have consistently shown that the effect of curcumin is highly dependent on its concentration [51,52]. It exhibits toxicity towards all types of cells within a specific concentration range; at lower concentrations, it seems to have no visible effect on cells. At a concentration of 25 μg/mL, no noticeable drastic changes were observed in cell ultrastructure. However, curcumin at high concentration led to changes in cell morphology and ultrastructure. At ½ MIC concentration, we observed a prominent electron-dense fibrillar layer on the cell wall surface. This visible extracellular matrix (ECM) layer has been previously described by Beauvais et al. [53], particularly in a yeast culture exhibiting flocculation. In the YPGal medium, the KV210 cells exhibited overexpression of the *FLO1* gene, resulting in strong flocculation. The ECM mainly consists of glucose, and a (1–6) branched (1–2)(1–3) mannan is synthesized in response to the adhesin Flo1p expression. The ECM is directly linked to the *FLO1* gene expression but is not crucial for flocculation [53]. Our study observed flocculation in response to curcumin, as evidenced by a decrease in the cell count in suspension form. Consequently, the beer produced with curcumin (B/CUR) exhibited increased clarity compared to the control beer. This flocculation effect was more pronounced at higher concentrations of curcumin. These findings collectively support the notion that curcumin’s effects on yeast growth are dose-dependent and highlight its potential to influence the growth dynamics of various yeast strains. A preceding study indicated that higher ethanol content had a more pronounced impact on the integrity of the cellular membrane in contrast to lower ethanol content. Ethanol levels of ≥10% were observed to enhance the fluidity of plasma membranes, disrupting cell membrane integrity [54]. Average cell volume of *S. cerevisiae* cultured in YPD medium notably decreased when exposed to 7.5% and 10% ethanol after a 16 h incubation period. Furthermore, the integrity of the cell membrane was compromised under the influence of ethanol stress. The researchers noticed the swelling or distortion of mitochondria and the presence of a single, large vacuole, both correlated with adding ethanol [55].

Beer is a beverage abundant in polyphenols and bioactive compounds, mainly because of the raw materials employed in brewing, including barley malt and hops. These ingredients contribute significantly to the beer’s overall content of beneficial compounds [56,57]. Beers with elevated phenolic and antioxidant levels exhibit superior quality with more stable sensory attributes, including enhanced flavor and aroma. Additionally, they have improved foam stability and a longer shelf-life compared to beers with lower antioxidant activity [58]. Based on our findings, the addition of curcumin in brewing wort led to a significant increase in both TPC and TFC content of the beer after fermentation and bottle aging. Several studies have aimed to enhance the functional properties of beer by augmenting its antioxidant activity by incorporating additives. In a study by Ulla et al. [59], they investigated the impact of adding an ethanolic extract of propolis (EEP) to beer at various concentrations (0.05, 0.15, and 0.25 g/L). The addition of EEP resulted in a linear increase in TPC, with values of 4.5%, 16.7%, and 26.7% higher than the control. Similarly, the TFC also showed a similar increase, with values 16.0%, 49.7%, and 59.2% above the control. In a separate study conducted by Seraa et al. [60], a commercial Hefeweizen beer was supplemented with different concentrations (1, 5, and 10 mg/mL) of grape skin extract (GSE). The addition of various GSE concentrations significantly increased the TPC and TFC content of the beer samples, ranging from 3.167 to 4.477 mg GAE/mL and from 0.841 to 1.226 mg CE/mL, respectively.

## 5. Conclusions

In conclusion, our study presents a novel exploration into the supplementation of beer with curcumin and its consequential impact on *S. boulardii* biofilm formation. Through a comprehensive analysis of fermentation performance, biofilm development, and gene expression, we have demonstrated that curcumin exerts a notable influence on the behavior of *S. boulardii* during the brewing process. While the initial addition of curcumin appeared to temporarily affect growth and fermentation performance, these effects became less prominent as fermentation progressed, ultimately leading to comparable outcomes in terms of growth and fermentation efficiency. Furthermore, our study revealed the role of curcumin in influencing biofilm formation. The reduction in biofilm biomass and extracellular polysaccharide content in the presence of curcumin points toward its potential as an agent for mitigating biofilm formation. Notably, the downregulation of specific *FLO* genes, except *FLO11*, suggests that curcumin’s influence on biofilm might be linked to the modulation of these gene expressions. As the beer industry continues to seek innovative approaches to enhance product quality, our study underscores the potential of curcumin as a biofilm-modulating agent, warranting further investigation and application in brewing practices.

## Figures and Tables

**Figure 1 biomolecules-13-01367-f001:**
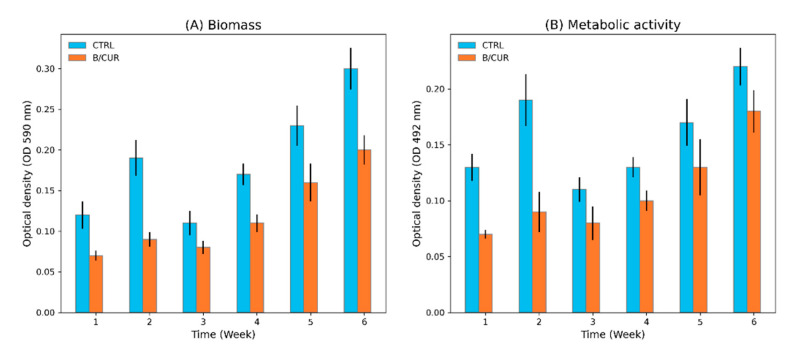
The mean absorbance values (±SD) of crystal violet (**A**) and XTT solutions (**B**) measured for the biofilm of *S. boulardii* on glass surface in control beer and B/CUR. A significant reduction in the mean CV and XTT values were observed in all time points (*p* < 0.05). Con—control beer; B/CUR—beer supplemented with curcumin.

**Figure 2 biomolecules-13-01367-f002:**
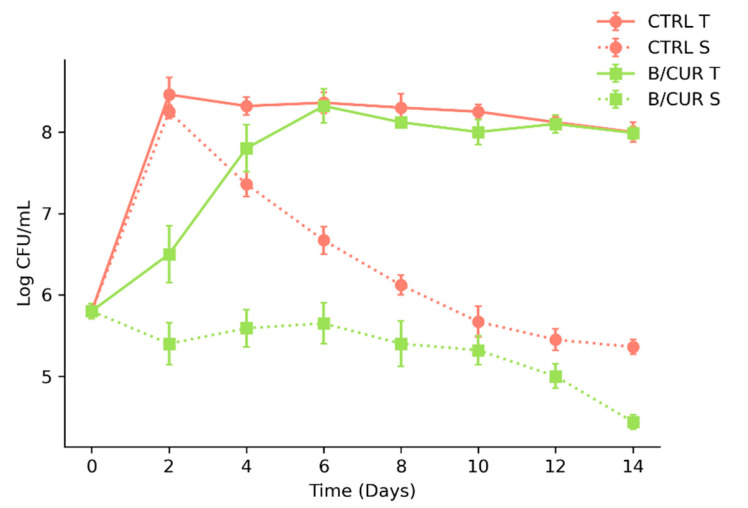
The total number of *S. boulardii* cells (T) and the number of *S. boulardii* cells in suspension (S) throughout the wort fermentation. Con—control beer; B/CUR—beer supplemented with curcumin. Each point represents the means of triplicate experiments ± standard deviation.

**Figure 3 biomolecules-13-01367-f003:**
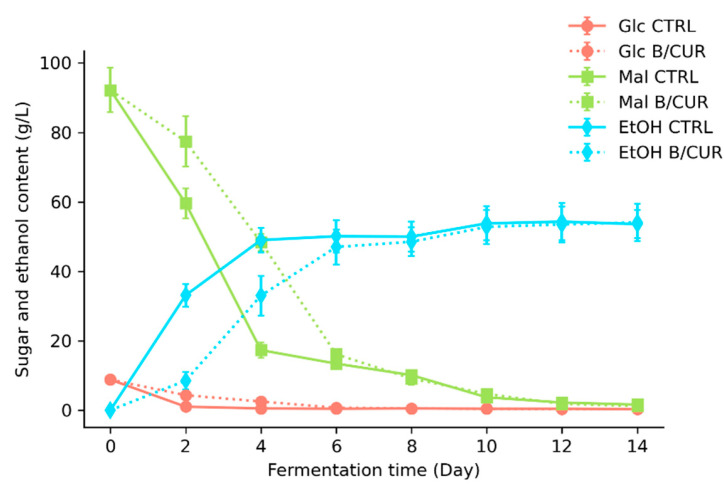
Uptake order of wort sugars and ethanol production by *S. boulardii* in control beer and B/CUR.

**Figure 4 biomolecules-13-01367-f004:**
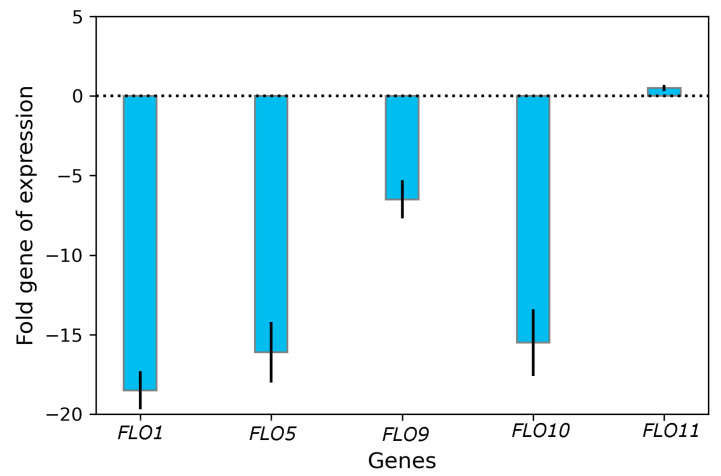
Fold change in *FLO* gene expression level of the B/CUR relative to control beer.

**Figure 5 biomolecules-13-01367-f005:**
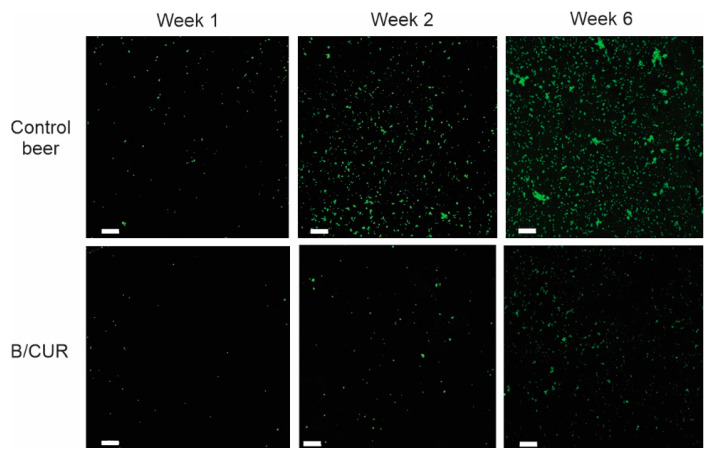
FITC-labeled concanavalin A staining of *S. boulardii* biofilm formed on the glass surface during the fermentation (weeks 1 and 2) and the last week of bottle aging (week 6). Measurements were made using a 10× objective lens. The scale bar represents 100 μm.

**Figure 6 biomolecules-13-01367-f006:**
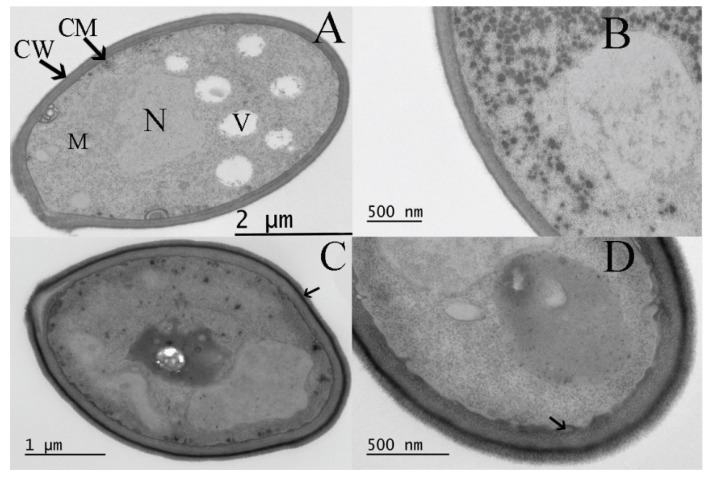
Transmission electron microscopy images of planktonic *S. boulardii* cells exposed to curcumin at concentration of 25 and 100 μg/mL (½ MIC) for 24 h. (**A**,**B**) Cells treated with 25 μg/mL showed an integral cell membrane (CM) and cell wall (CW) structures and a homogeneous cytoplasm; M, mitochondria; N, nucleus and V, vacuole. (**C**,**D**) *S. boulardii* stressed by high concentration of curcumin (100 μg/mL) showed thickened cell wall.

**Table 1 biomolecules-13-01367-t001:** Primers used for quantitative real-time PCR.

Gene	Forward Primer Sequence (5′–3′)	Reverse Primer Sequence (5′–3′)
*FLO1*	ACAGAGACAACAAAGCAAACC	ACACACACCAGATTCGCAG
*FLO5*	ACCCCAACAAACGTAACCCT	GGTGCTAGCTGTTGTTGGAG
*FLO8*	CGTATCCAGATTCAATTCCTCC	GCTGTTCACTATTCGTTGCC
*FLO9*	ACAACAGAGCAAACCACAG	ACCGTAACAACATCATTCACAG
*FLO10*	CTACACAACACCCACCAAC	AAGGCAACATTACCTCCAAC
*FLO11*	AACCAAGTCCATCCCAACC	GCGAGTAGCAACCACATAAAG
*18S*	TTAATGACCCACTCGGCAC	CACCACCACCCACAAAATC

**Table 2 biomolecules-13-01367-t002:** Results obtained for bioactive compounds (TPC and TFC) for control beer and B/CUR. Values marked with the same letter do not show significant differences (*p* < 0.05).

	Fermentation		Bottle Aging	
Parameter	Control	B/CUR	Control	B/CUR
pH	4.57 ^a^	4.74 ^b^	4.52 ^a^	4.32 ^c^
TPC (mg GAE/L)	229 ± 18 ^a^	283 ± 22 ^b^	209 ± 15 ^a^	261 ± 24 ^b^
TFC (mg QE/L)	15.2 ± 2.6 ^a^	23.6 ± 2 ^b^	12.4 ± 3.3 ^a^	19.1 ± 2.5 ^b^

**Table 3 biomolecules-13-01367-t003:** The consumer acceptance ratings for the sensory characteristics and overall impression of beer made with and without curcumin. Values indicated by the same letter do no show significant differences (*p* < 0.05).

Attributes	B/CUR	Control
Appearance	8.3 ± 0.48 ^a^	7.4 ± 0.48 ^b^
Aroma	7.0 ± 0.90 ^a^	7.1 ± 0.67 ^a^
Taste	7.7 ± 0.76 ^a^	7.1 ± 0.83 ^b^
Mouthfeel	6.8 ± 0.85 ^a^	6.5 ± 0.61 ^a^
Aftertaste and finish	7.3 ± 0.89 ^a^	6.61 ± 0.77 ^b^
Overall impression	8.1 ± 0.67 ^a^	7.6 ± 0.69 ^b^

## Data Availability

The data presented in this study are available in this article.

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
