# Peer review of "Antibiofilm Effect of Curcumin on Saccharomyces boulardii during Beer Fermentation and Bottle Aging"

_biomolecules, 2023, doi:10.3390/biom13091367_

Round 1

Reviewer 1 Report

The authors provide  interesting study on the anti-biofilm effect of curcumin on Saccharomyces boulardii during beer fermentation and bottle aging. The manuscript is well written, but some minor mistakes can be addressed, which can improve their work. Therefore, I recommend that the authors revise their manuscript according to the comments below.

#1 Adds quantitative relevant informations to the abstract. I recommend authors use the following reference to adjust their abstract (https://doi.org/10.1016/j.carbon.2007.07.009).

#2 The novelty of the work is not expressed explicate in a full way. In which aspect this work is original and better than others?

3# The scales shown in Fig. 5 are not clear.

#4 The visual representation in the graphics appears less professional. Employing a contemporary tool like Origin software could enhance the quality and suitability of the visuals.

#5 Please improve your conclusion. The main results were not highlighted.

#6 Please replace old references by latest ones. As can be seen only 15 of 57 references are listed as new references (2018-2023), which is unsuitable. At least 75% of the references of a modern manuscript should be between 2018-2023. Check your references.

Author Response

Dear Reviewer 1,

I truly appreciate the attention you've given to every detail and the constructive feedback that has undoubtedly contributed to enhancing the overall impact of the manuscript. Please find below you comments and our responses.

Comment

#1 Adds quantitative relevant informations to the abstract. I recommend authors use the following reference to adjust their abstract (https://doi.org/10.1016/j.carbon.2007.07.009).

Response

The introduced reference was useful not only for adjusting the current manuscript but also for future publications. We deleted the first beginning sentence without losing the information and made it shorter to get straight to the point.

#2 The novelty of the work is not expressed explicate in a full way. In which aspect this work is original and better than others?

Response

We highlighted the novelty of this work in "conclusion"

3# The scales shown in Fig. 5 are not clear.

Response

We increased the thickness of the scale bars in Figure 5 to be visible.

#4 The visual representation in the graphics appears less professional. Employing a contemporary tool like Origin software could enhance the quality and suitability of the visuals.

Response

Thanks for your attention. The previous figures were extracted in Excel software. Now, the quality of graphics is improved.

#5 Please improve your conclusion. The main results were not highlighted.

Response

We revised the conclusion to highlight the results and novelty of the study.

#6 Please replace old references by latest ones. As can be seen only 15 of 57 references are listed as new references (2018-2023), which is unsuitable. At least 75% of the references of a modern manuscript should be between 2018-2023. Check your references.

Response

We updated the references.

Reviewer 2 Report

The manuscript entitled "Anti-biofilm effect of curcumin on Saccharomyces boulardii during ‎beer fermentation and bottle aging" studies the effect of curcumin on the yeast S. boulardii during beer production. The authors considers curcumin a stressing factor on S. boulardii, but despite this the authors suggests that curcumin can be added during beer fermentation, because it can prevent biofilm production, and it can positively influence the organoleptic properties of the beer.

 The manuscript is very well written. The aim appears clear, and it is supported by the obtained results.

 However, some minor mistakes appears and have to be addressed.

 Line 49. Please change S. cerevisiae in Saccharomyces cerevisiae, since it’s the first time that the authors mention the name of the yeast.

 Line 86: please change the sentence “was as following steps” with “was as follows”

  ml or mL? Please make uniform throughout the text.

 Line 100. Is CUR an abbreviation of curcumin? Please specify and make uniform throughout the text.

 Table 1: Did the authors design the primers? Specify it, if this is the case, otherwise write the reference.

 Line 185 please change “performed triplicate” in “performed in triplicate”

 Line 183-188. Please specify the qRT-PRC cycle used by the authors

 Line 206. Please change 4 C in 4 °C.

 Figure caption. This is just a suggestion. The authors write the acronym of the control beer, as “Con”, maybe it would be easier to understand an acronym such as CTRL

 Figure 4. Please verify the ordinate axis legend.

 Line 340. Figure 4 or figure 5?

 Reference 9. Please delete the sentence in caps lock.

Author Response

Dear Reviewer 2

I am so grateful for your valuable time, thoughtful insights, and meticulous evaluation of my manuscript. Your comments and suggestions have been instrumental in refining the quality and clarity of the content. Please find below our response to your comments.

Line 49. Please change S. cerevisiae in Saccharomyces cerevisiae, since it’s the first time that the authors mention the name of the yeast.

Response

  1. cerevisiae in Line 49 was changed to Saccharomyces cerevisiae

Line 86: please change the sentence “was as following steps” with “was as follows”

Response

the sentence “was as following steps” changed to “was as follows”

ml or mL? Please make uniform throughout the text.

Response

Throughout the text ml changed to mL

Line 100. Is CUR an abbreviation of curcumin? Please specify and make uniform throughout the text.

Response

Thanks for your attention. We did not use an abbreviation for curcumin, so "CUR" in lines 100 and 230 was changed to "curcumin".

Table 1: Did the authors design the primers? Specify it, if this is the case, otherwise write the reference.

Response

The primers were designed by authors. We mentioned it in line 178.

Line 185 please change “performed triplicate” in “performed in triplicate”

Response

Thanks for your attention. The sentence was changed.

Line 183-188. Please specify the qRT-PRC cycle used by the authors

Response

The reaction set-up and qRT-PCR cycle are added in lines 182-188.

Line 206. Please change 4 C in 4 °C.

response

the degree sign was added

Figure caption. This is just a suggestion. The authors write the acronym of the control beer, as “Con”, maybe it would be easier to understand an acronym such as CTRL

Response

Thanks for your suggestion. We used the acronym “CTRL” instead of “Con”

Figure 4. Please verify the ordinate axis legend.

Response

the ordinate axis legend‎ was verified

Line 340. Figure 4 or figure 5?

Response

Figure number was corrected.

Reference 9. Please delete the sentence in caps lock.

Response

We updated reference 9.
